# Localized Controlled Release of Kynurenic Acid Encapsulated in Synthetic Polymer Reduces Implant—Induced Dermal Fibrosis

**DOI:** 10.3390/pharmaceutics14081546

**Published:** 2022-07-25

**Authors:** Layla Nabai, Aziz Ghahary, John Jackson

**Affiliations:** 1BC Professional Fire Fighters’ Burn & Wound Healing Research Lab, ICORD, The Blusson Spinal Cord Centre, 818 West 10th Ave, Vancouver, BC V5Z 1M9, Canada; layla1@mail.ubc.ca (L.N.); ghaharya@yahoo.com (A.G.); 2Faculty of Pharmaceutical Sciences, The University of British Columbia, 2045 Westbrook Mall, Vancouver, BC V6T 1Z3, Canada

**Keywords:** kynurenic acid, PLGA, microsphere, fibrosis

## Abstract

Excessive fibrosis following surgical procedures is a challenging condition with serious consequences and no effective preventive or therapeutic option. Our group has previously shown the anti-fibrotic effect of kynurenic acid (KynA) in vitro and as topical cream formulations or nanofiber dressings in open wounds. Here, we hypothesized that the implantation of a controlled release drug delivery system loaded with KynA in a wound bed can prevent fibrosis in a closed wound. Poly (lactic-co-glycolic acid) (PLGA), and a diblock copolymer, methoxy polyethylene glycol-block-poly (D, L-lactide) (MePEG-b-PDLLA), were used for the fabrication of microspheres which were evaluated for their characteristics, encapsulation efficiency, in vitro release profile, and in vivo efficacy for reduction of fibrosis. The optimized formulation exhibited high encapsulation efficiency (>80%), low initial burst release (~10%), and a delayed, gradual release of KynA. In vivo evaluation of the fabricated microspheres in the PVA model of wound healing revealed that KynA microspheres effectively reduced collagen deposition inside and around PVA sponges and α-smooth muscle actin expression after 66 days. Our results showed that KynA can be efficiently encapsulated in PLGA microspheres and its controlled release in vivo reduces fibrotic tissue formation, suggesting a novel therapeutic option for the prevention or treatment of post-surgical fibrosis.

## 1. Introduction

Fibrosis is a common complication, often with serious consequences, affecting millions of people around the world undergoing surgical procedures annually [1,2,3,4,5]. According to international studies, 5% to 33% of patients undergoing surgery for lumbar intervertebral herniation are likely to develop epidural fibrosis (EF) [6], which has been implicated in the etiology of persistent pain after back surgery [7]. Postoperative adhesions, the leading cause of intestinal obstruction, occur after almost every abdominal surgery [8]. Another frequent post-operative complication is adhesion-related chronic abdominal and pelvic pain, which affects 20–40% of patients who have undergone surgery on the female genital or alimentary tract, and has an enormous impact on patients’ quality of life [9,10]. Arthrofibrosis, an abnormal scarring, develops in 3–10% of cases after total knee arthroplasty (TKA) in patients with degenerative knee joint diseases [11]. Implantation of biomedical devices is an important part of modern medical care. Despite the inert and non-toxic composition of biomedical devices, a variety of adverse reactions, including inflammation, fibrosis, coagulation, and infection can be induced by their implantation [12]. Following the inflammatory reaction, the fibroblasts migrate to the surface of the implant and subsequently differentiate into myofibroblasts. The collagen deposited by these cells forms a capsule that surrounds the device and separates it from the surrounding soft tissues [13].

The inflammation and capsule formation around the biomedical implants such as drug delivery systems, electrodes placed in neuronal tissues, and glaucoma implants, may interfere with the proper function of that device [14,15,16,17]. Additionally, capsular contracture may occur and cause distortion and firmness of the originally soft devices [13]. Implant-based breast reconstruction and augmentation are commonly performed reconstructive and aesthetic procedures in Western countries [18,19]. According to The American Society for Aesthetic Plastic Surgery, the number of breast augmentation procedures in the United States has increased by 207% in the last two decades [19]. One of the most unfavorable complications of breast augmentation and reconstruction is the capsular contracture that originates from pathological fibrosis and occurs in 1.3–30% of patients, who must often undergo revision surgery to surgically resect the scar tissue and replace the implant [20].

Despite several strategies such as meticulous surgical techniques to reduce tissue damage [21] or using non-absorbable and bioabsorbable barriers [22,23,24], and advances in wound care, prevention of post-surgical fibrosis remains challenging. Currently recommended local treatments are usually painful, time-consuming, or expensive [25], and systemic treatments have limitations, such as, variable availability of the drug at the site of injury, failure to achieve or maintain adequate concentrations for the duration of healing, and systemic toxicity [26,27]. Our group has previously shown that the metabolites of L-tryptophan, Kynurenine (Kyn) and Kynurenic acid (KynA), have anti-fibrotic effects through reduction of fibroblast proliferation, type-1 collagen, and fibronectin expression, and increased expression of MMP [28]. Furthermore, topical cream formulations of the metabolites and nanofiber dressings loaded with KynA have been successfully tested in animal models with open wounds [28,29]. However, most surgical wounds are closed with sutures and the fibrotic complications usually appear weeks or months after the closure of the wound. Controlled release drug delivery systems have been developed to overcome the obstacle of delivering a therapeutic agent locally at the effective concentration for the appropriate time [30,31,32,33]. Biocompatible, biodegradable polymeric microspheres are one of the most used drug delivery systems. Poly (lactic-co-glycolic acid) (PLGA) is an FDA and European Medicine Agency-approved polymer for controlled drug delivery uses even through parenteral administration [34]. The commercially available PLGA, with different molecular weights (MWs) and compositions, may be processed into almost any shape and size, with variable degradation rates [35]. However, the rate of hydrophobic drug release from PLGA only microspheres is extremely slow and the addition of diblock copolymers such as methoxy polyethylene glycol-block-poly (D, L-lactide) (MePEG-b-PDLLA) to the formulation allows modulation of drug release [36].

In this study, we hypothesized that controlled delivery of KynA, encapsulated in PLGA microspheres, may reduce implant-induced fibrosis. To address this hypothesis, we investigated the encapsulation of KynA in microspheres fabricated from PLGA and tested the anti-fibrotic effect of fabricated microspheres in an animal model.

## 2. Materials and Methods

### 2.1. Materials

KynA (Sigma-Aldrich, St. Louis, MO, USA), PLGA (85/15, inherent visc. (IV) = 0.61 dL/g inCHCl3 at 30 °C) (Birmingham Polymers, Birmingham, AL, USA), poly (vinyl alcohol) (PVA) (98% hydrolyzed, MW 25,000) (Polysciences, Warrington, PA, USA) were used as supplied. The diblock copolymer, methoxy polyethylene glycol-block-poly (D, L-lactide) (MePEG-b-PDLLA) (60:40 *w*/*w*) was synthesized as previously described [36]. The MePEG molecular weight was 2000 Da and the total molecular weight of the diblock (measured by GPC) was 3500 Da. Sodium chloride, acetonitrile, methanol (HPLC grade), dichloromethane (DCM), and Triethylamine (TEA) were from Fisher (Fair Lawn, NJ, USA), sodium phosphate monobasic (EMD chemicals, Gibbstown, NJ, USA). Trichrome stain (Masson) (HT15-1KT) and Weigert’s Iron Hematoxyline set (HT1079), trans-4-hydroxy-L-proline, were bought from Sigma (St. Louis, MO, USA). Phenylisothiocyanate (PITC, Edman’s Reagent), proteinase K, and 4′,6-diamidino-2-phenylindole (DAPI) were purchased from Thermo Scientific (Rockford, IL, USA).

Rabbit monoclonal antibody (E 184) to alpha smooth muscle actin (ab32575) was obtained from Abcam Inc. (Cambridge, MA, USA) as the primary antibody and the secondary antibody, biotinylated goat anti-rabbit antibody, DAB substrate kit for peroxidase SK-4100 and Peroxidase standard pk4000 were from Vector Laboratories (Burlington, ON, Canada).

NanoDrop spectrophotometer (NanoDrop Technologies, Wilmington, DE, USA) was used to measure RNA concentration and Applied Biosystems^®^ 7500 Fast Real-Time PCR System was used for qPCR. Superscript II First Strand cDNA Synthesis kit was from Invitrogen and SYBR^®^ Green PCR Master-Mix kit was purchased from Applied Biosystems, Warrington, UK.

Rat MMP-13 forward (5′-TTGTTGCTGCCCATGAGCTT-3′) and reverse (5′-ACTTTGTCGCCAATTCCAGG-3′), rat Col-1α1 forward (5′-CAAGAATGGCGACCGTGGT-3′) and reverse (5′-GGTGTGACTCGTGGAGCCA-3′), rat α-SMA forward (5′-ACTGGGACGACATGGAAAAG-3′) and reverse (5′-CATCTCCAGAGTCCAGCAGA-3′) primers were used. Rat β-actin forward (5′-TATCGGCAATGAGCGGTTCC-3′) and reverse (5′-GTGTTGGCATAGAGGTCTTTACG-3′) primers were used as the reference gene for qPCR.

An overhead stirrer (BDC 2002 Caframo, Wiarton, ON, Canada) was used for the fabrication of microspheres. Polarized light microscope and scanning electron microscopy (Hitachi S-3000 N scanning electron microscope) were used to examine the morphology of the microspheres. A Waters Acquity HPLC system with UV detection and a Novapak^®^ C18 column and Pro E software (Empower 3, 2010) (Waters, Mississauga, ON, Canada) was used for the quantification of kynurenic acid. Pre-cut polyvinyl alcohol (PVA) sponges, 1 cm in diameter, were kindly provided by Medtronic, Inc. (Jacksonville, FL, USA).

### 2.2. Fabrication of KynA Loaded PLGA Microspheres

To fabricate controlled release microspheres, we chose to use PLGA with a ratio of LA to GA of 85:15. The low solubility of KynA in water (0.9% at 100 °C) allowed us to use single Oil in Water emulsion/solvent evaporation technique for the preparation of PLGA microspheres. For 5% KynA-loaded microspheres, 25 mg of KynA as the powder was added to a solution of 475 mg of PLGA 85/15 in 1.5 mL of dichloromethane (DCM), vortexed for 30 s, and incubated at 60 °C for 5 min. Then the PLGA-KynA mixture was slowly pipetted into a 100 mL aqueous solution of PVA 2.5% (*w*/*v*) while stirring at 600 rpm with an overhead stirrer at room temperature, where the propeller was fitted tightly in the glass container. After 10 min, the stirring speed was reduced to 450 rpm and continued up to 2 h. The microspheres were separated by gravity, washed 3 times with distilled water, and air dried overnight. In some studies, an amphiphilic excipient, MePEG-PDLLA diblock copolymer, was added to the polymer solution in DCM with a ratio of 10, 17, and 20% *w*/*w* to PLGA.

Polymer-only microspheres (empty microspheres) were prepared using the same procedure.

### 2.3. Characterization of the Fabricated Microspheres

The morphology of empty and KynA loaded PLGA or PLGA/diblock microspheres were examined using a Polarized light microscope and scanning electron microscopy, after coating the samples with a thin layer of gold under vacuum. The particle size analysis of the microspheres was performed using Malvern Hydro 2000 laser diffraction particle size analyzer as previously described [37]. Three measurements were performed on each batch of four separate preparations of microspheres and the particle size was expressed as the volume weighted mean.

### 2.4. Measurement of KynA Using HPLC

For calibration, standard solutions of kynurenic acid in distilled water at concentrations ranging from 50 µg/mL down to 0.024 µg/mL were prepared. The mobile phase was methanol:10 mM sodium phosphate monobasic, pH: 2.8 (27:73 *v/v*) as previously described by Badawy A. et al. [38] with a flow rate of 1 mL/min and 20 μL injection volume. The system, a Waters 1525 binary pump with UV/visible detector and reverse-phase column, was run isocratically and the KynA absorbance peak was monitored at 330 nm.

### 2.5. Evaluation of the Encapsulation Efficiency

The encapsulated KynA in PLGA or PLGA/diblock microspheres was calculated by measuring the amount of encapsulated KynA, after extraction from microspheres, multiplied by extraction efficiency.

The extraction efficiency was determined by calculating the recovery rate of the known amount of drug from the dried film. A mixture of PLGA: KynA (95:5) was dissolved in DCM with a final concentration of KynA being 1 mg/mL of DCM. Then 200 µL of the solution in four separate tubes was dried under nitrogen flow. After adding one mL of DCM and 9 mL of dH_2_O to each tube containing dried film, tightly closed tubes were tumbled for one hour at room temperature. The concentration of KynA in the water phase was measured using HPLC.

For extraction of KynA from microspheres, approximately 5 mg of microsphere were dissolved in 1 mL of DCM in quadruplicate and followed the procedure as mentioned for the dried film.

The encapsulation efficiency (%) was expressed as: (the amount of KynA in the microspheres/the theoretical amount of KynA in the microspheres) × 100.

### 2.6. KynA Release Profile

For in vitro release profile of KynA from PLGA or PLGA/diblock microspheres 5 mg of different formulations were weighed into separate tubes in quadruplicate. After adding 10 mL phosphate-buffered saline (PBS) solution (10 mM, pH 7.4) to each tube, tightly closed tubes were incubated at 37 °C while rotating. The release medium was collected at 1, 2, 3, 7, 15 days and replaced by fresh buffer. The concentration of released KynA was measured using HPLC. In another set of experiments, the extended-release profile of the selected formulation was obtained as mentioned above and the release medium was collected at 1, 2, 3, 6, and then every 5–6 days up to 70 days.

### 2.7. Stability of the KynA after Encapsulation

To determine the stability of KynA during the process of fabrication of microspheres and following the release, KynA loaded PLGA + 17% MePEG-diblock microspheres were incubated in PBS for 48 h as described above. The UV absorbance at 330 nm and the retention time (t_R_) of the released KynA was compared to the freshly made, standard KynA solution.

### 2.8. In Vivo Animal Studies

Pre-cut, 1 cm in diameter polyvinyl alcohol (PVA) sponges (kindly provided by Medtronic, Inc., Jacksonville, FL, USA) were hydrated and sterilized in PBS. The required amount of KynA microspheres was calculated based on the previously published in vitro results [28], the encapsulation efficiency, and estimated release profile. Equal quantities of empty microspheres were also weighed and loaded separately in the dead space of the PVA sponges. Eight male Sprague-Dawley rats aged 5 weeks were anesthetized using isoflurane inhalation and six full-thickness 1 cm long incisions were made on the back of each rat. Incisions were divided into three groups: (i) PVA sponge implant; (ii) PVA sponge loaded with KynA microspheres; and (iii) PVA sponge loaded with empty microspheres. Following subcutaneous implantation in a pouch made under the panniculus carnosus, wounds were sutured and dressed according to the protocol. Rats were euthanized at 35- and 66-days post-implantation, and PVA sponges + overlying skin were harvested. One set of the PVA sponges from each rat was cut in half, one half was fixed in 10% neutral buffered formaldehyde, dehydrated, embedded in paraffin, used for histology and the other half was stored at −80 °C for RNA extraction. The second set was stored at −80 °C for the hydroxyproline assay.

### 2.9. In Vivo Release of KynA

Any residual microspheres collected from samples used for hydroxyproline assay were rinsed with dH_2_O, air dried, weighed, and the remaining encapsulated KynA was extracted and measured as described above.

### 2.10. Histological Analysis of Fibrosis

For histological evaluation, paraffin-embedded tissue sections (5 μm) were rehydrated and stained according to the standard H&E and Masson’s trichrome staining protocol [39].

### 2.11. In Vivo Biological Activity; Quantification of Collagen

The amount of collagen deposited inside the second set of PVA sponges was quantified using a hydroxyproline assay. Briefly, lyophilized PVA sponges were cut into small pieces and incubated with one mL of proteinase K (in Tris-HCl buffer) at 54 °C while shaking for 5–6 h. After centrifugation and removal of the PVA pieces and remaining microspheres, the aliquots of the resultant homogenates were incubated with an equal volume of 12 N HCl at 105 °C overnight to hydrolyze the collagen. Dried samples were suspended in 40 µL of ethanol: dH_2_O: TEA (2:2:1) and dried again. Derivatization to phenylthiocarbamyl form was achieved by adding 40 µL of ethanol: dH_2_O: TEA: PITC (7:1:1:1) to each sample and incubating at room temperature for 20 min before drying. After resuspension in one mL of analysis solution (dH_2_O: Acetonitrile, 7:2) and centrifugation samples were analyzed by HPLC. One mg of purified type I bovine collagen was hydrolyzed and derivatized as a control.

For the creation of a calibration curve, different amounts of hydroxyproline (1–320 µg) were dried from freshly prepared stock solutions of trans-4-hydroxy-L-proline in dH_2_O and derivatized in the same manner as the samples before analysis.

Hydroxyproline eluted isocratically at 23 °C using a mobile phase of 94% 140 mM sodium acetate, 0.05% TEA, and 6% Acetonitrile, pH 6.4, as eluent at 1 mL/min. UV absorption was monitored at 260 nm.

### 2.12. Total Tissue Cellularity

To determine total tissue cellularity, 5 μm, paraffin-embedded tissue sections were stained with 4′,6-diamidino-2-phenylindole (DAPI). For each sample, 30 fields of PVA dead space were photographed under ×200 magnification using a Zeiss fluorescent microscope and Axiovision software. The photographs were coded and the number of nuclei in 10 randomly selected fields, for each sample, was counted by two independent individuals using Image Pro Plus 4.5 software (Media Cybernetics, Inc., Rockville, MD, USA). The counts from the corresponded fields were averaged.

### 2.13. In Vivo Biological Activity; α-SMA Protein Expression

Formalin-fixed, paraffin-embedded tissue sections were stained for α-SMA expression as described in Abcam IHC-paraffin protocol with some modifications.

Briefly, after deparaffinization, re-hydration, and heat-induced epitope retrieval with sodium citrate buffer (10 mM, pH 6.0) + 0.05% Tween 20, sections were washed with tris-buffered saline (TBS) (20 mM, pH 7.4) + 0.025% triton X-100, permeabilized with 0.2% triton X-100, and blocked with 5% normal goat serum + 5% bovine serum albumin in TBS. Following overnight incubation with rabbit monoclonal antibody to α-SMA at 4°C, sections were washed and incubated with 3% H_2_O_2_ before adding the biotinylated goat anti-rabbit antibody. After washing, sections were incubated with a mixture of avidin-biotin from the Vectastain ABC kit, and the colored product of the enzyme horseradish peroxidase (HRP) was developed with the DAB substrate kit for peroxidase. The nuclei were counterstained with hematoxylin.

### 2.14. In Vivo Biological Activity; MMP-13, Type-1 Collagen, and α-SMA Gene Expression

The presence of MMP-1 in rats is controversial and MMP-13 is the main collagenase implicated in the degradation of fibrillar collagen in rats [40,41]. As such, the level of MMP-13, type-I collagen, and α-SMA expression in tissue samples was examined by RNA extraction using Trizol reagent according to the manufacturer’s instructions (Invitrogen). Equal concentrations of RNA samples were then reverse transcribed to cDNA using a Superscript II First Strand cDNA Synthesis kit. qPCR was performed using the SYBR^®^ Green PCR Master-Mix kit. The changes in gene expression were normalized to the results of the PVA alone group.

### 2.15. Statistical Analysis

Data were expressed as mean ± SD of three or more independent experiments unless otherwise indicated. One-way analysis of variance (ANOVA) followed by Scheffe post hoc test was used to calculate statistical significance. *p* values < 0.05 were considered statistically significant in this study.

## 3. Results

### 3.1. Morphology and Size Distribution of the Fabricated Microspheres

Scanning electron microscopy images of PLGA or PLGA+ 10, 17, and 20%MePEG-diblock microspheres loaded with 5% KynA showed similar morphology (Figure 1A).

The size distribution of the PLGA (507.47 ± 31.92 μm), PLGA + 10% MePEG-diblock (507.51 ± 32.61 μm), PLGA + 17% MePEG-diblock (451.71 ± 59.28 μm), and PLGA + 20% MePEG-diblock (473.85 ± 26.89 μm) microspheres exhibited no statistically significant difference (Figure 1B).

### 3.2. Encapsulation Efficiency, In Vitro Release Profile, and Stability of KynA

Polarized light microscopy images revealed successful KynA encapsulation for all four formulations (Figure 2A). Quantitative analysis of the encapsulation efficiency of different formulations revealed more than 80% encapsulation efficiency for PLGA, PLGA + 10%, and PLGA + 17% MePEG-diblock, while PLGA + 20% MePEG-diblock provided the lowest (63%) encapsulation efficiency (Figure 2B).

The cumulative released KynA from PLGA + 0, 10, 17, and 20% MePEG-diblock microspheres at 1, 2, 3, 7, 15 days revealed that less than 2% of the encapsulated KynA was released from PLGA+ 0 and 10% MePEG-diblock microspheres in the first 24 h and only 4.5% was released up to 15 days (Figure 2C). On the other hand, the burst release on the first day for PLGA+ 20% MePEG-diblock was 45% and almost 57% of the encapsulated KynA was released up to 15 days. In comparison to the other formulations, PLGA +17% MePEG-diblock had a release profile with 10% burst release followed by a gradual release of up to 15% in 15 days.

As indicated in Figure 2, the peak absorbance and the retention time (t_R_) of the freshly made standard solution of the KynA (Figure 2D(a)) and the KynA released from fabricated microspheres (Figure 2D(b)) were identical. Since the retention time measured under particular conditions is considered an identifying characteristic of a given analyte, the identical results confirm the stability of KynA in the process of fabrication and following the release from microspheres.

### 3.3. Morphology of the Polymer Only and Extended-Release Kinetics of KynA Loaded Microspheres In Vitro and In Vivo

Based on the initial short-term release kinetics, the PLGA + 17% MePEG-diblock formulation was chosen for further studies. Scanning electron microscopy images of the empty microspheres (a,c) and KynA loaded microspheres (b,d), using the selected formulation, are shown in Figure 3A.

The result of the extended-release profile of PLGA + 17% MePEG-diblock and the corresponding phases of wound healing are shown in Figure 3B. Following a low initial burst release (10%) a 5% release was observed in 2 weeks. After a lag period with almost no release for 20 days, the remaining drug was gradually released from day 36 up to day 70, with maximum release seen between day 42 to 48.

A comparison of the residual amount of KyA in microspheres retrieved from in vivo studies with that of in vitro revealed that the remaining amount of KynA inside the microspheres after 35 and 66 days is significantly higher in vitro than in vivo (83.5% ± 2.2 vs. 48.9% ± 4.3 and 31.7%± 4.1 vs. 2.3% ± 0.5, respectively) (*n* = 4, *p* < 0.05) (Figure 3C).

### 3.4. KynA Microspheres Decrease New Tissue Growth and Total Cellularity inside Implanted PVA Sponges

The H&E staining of the samples harvested after 35 days showed that almost the same amount of granulation tissue had grown inside the dead space of the PVA sponges in all three groups, while after 66 days, the fibrotic tissue in the dead space of the PVA + KynA microspheres was less compared to the PVA alone and PVA + empty microspheres (Figure 4A).

Quantification of the total tissue cellularity inside the dead space of the PVA sponges revealed that the number of cells in the PVA alone group, after 35 and 66 days, (174 ± 16, 171 ± 4.3 respectively) was significantly higher than the PVA + empty microspheres (148 ± 1.5, 155 ± 2.6 respectively) and PVA + KynA microspheres (143.3 ± 10, 144 ± 6.5 respectively) (Figure 4B,C). No significant difference was observed between PVA + empty microspheres and PVA + KynA microspheres.

### 3.5. KynA Microspheres Decrease Collagen Deposition inside and around PVA Sponges

Representative images of Masson’s trichrome staining of the sections harvested after 35 and 66 days are shown in Figure 5A. Quantitative analysis of collagen by hydroxyproline assay showed no significant difference between the three groups after 35 days (PVA alone 1.4 ± 0.2, PVA + empty mic. 1.8 ± 0.1, PVA + KynA mic. 1.2 ± 0.4 mg/PVA) (Figure 5B). However, the amount of collagen in PVA + KynA microspheres (0.3 ± 0.5 mg/PVA) was significantly less than PVA alone (6.7 ± 2.8 mg/PVA) and PVA + empty microspheres (2.7 ± 0.9 mg/PVA) (*n* = 4, *p* = 0.009) after 66 days. The difference between PVA and PVA + empty microspheres was also significant.

### 3.6. Col1α1, MMP-13, and α-SMA Expression In Vivo

As shown in Figure 6A, at day 35, there was no statistically significant difference in the expression of Col1α1, MMP-13, or α-SMA at the gene level among groups. However, the expression of α-SMA was significantly reduced in both PVA + empty microspheres and PVA + KynA microspheres (0.21 ± 0.2, 0.23 ± 0.34 respectively, *n* = 3, *p* = 0.009) at day 66 (Figure 6B). The changes in Col1α1 and MMP-13 gene expression at day 66 were not statistically significant. IHC staining of the sections revealed that few cells, in all three groups, express α-SMA at day 35 (Figure 6C). However, the number of cells positive for α-SMA increases in the PVA alone group at day 66, while few cells are positive in PVA + empty microspheres and PVA + KynA microspheres (Figure 6D).

## 4. Discussion

Controlled release of an anti-fibrotic agent from implantable, biodegradable, and biocompatible microspheres is a novel approach to address the limitations associated with current therapeutic options for the challenging condition of post-surgical fibrosis. In this study, we demonstrated that KynA is efficiently encapsulated in microspheres fabricated from the PLGA copolymer, without any change in its chemical properties. Furthermore, the addition of the MePEG diblock up to 17% to the PLGA did not affect the drug encapsulation efficiency, morphology, or size of the fabricated microspheres. However, a 3% increase in MePEG diblock to PLGA ratio from 17% up to 20% reduced the encapsulation efficiency by 20% from 83% down to 63%. Although it has been reported that the addition of polyethylene glycol (PEG) to polymeric formulations results in reduced encapsulation efficiency for drugs and proteins [35], Jackson et al. found no difference in encapsulation of paclitaxel following the inclusion of different concentrations of amphipathic diblock copolymers in PLGA solutions [36]. Further, we found that when the diblock ratio increased from 17% up to 20%, the initial burst release of the drug increased from 10% up to 45%, likely due to the release of the diblock from the PLGA matrix and a critical increase in the polymer porosity allowing penetration of water into the polymer matrix, followed by increased release of solubilized hydrophobic KynA within diblock copolymer micelles formed in pores, as previously shown for paclitaxel [36].

A high initial burst release of KynA not only has a detrimental impact on wound healing but also decrease the available amount of the drug for the prevention of fibrosis at later stages of healing. Therefore, based on the encapsulation efficiency and the low level of the initial burst release, the PLGA + 17% MePEG diblock formulation was selected for subsequent studies. The extended in vitro release kinetics of the selected formulation revealed a biphasic drug release separated with a relatively long “lag-time” (20 days) of almost no release, in line with previous studies [35,42]. These later increase in drug release may arise from the degradation of the PLGA 85:15 in water which is reported to occur after 5–6 weeks [43]. This release profile provides the anti-fibrotic agent during the remodeling phase of the wound healing, when the imbalance between collagen deposition and degradation results in excessive fibrotic tissue development. The residual drug analysis (Figure 3C) in the in vitro incubated microspheres closely matched the release profiles by accounting for most of the drug and reinforcing the stability of the drug over these extended time frames. The quantitative analysis of the remaining KynA in microspheres collected at two time points from the implanted sponges showed lower levels of residual drug in these microspheres compared to in vitro microspheres (Figure 3C). The changes in pH of the wound environment from basic to a neutral and then to an acidic state might contribute to the faster degradation of the polymer and drug release in vivo [44]. However, a significant amount of undegraded KynA remained at the wound site from 35 to 66 days suggesting that the in vitro and in vivo release profiles provided similar levels of active drug at various time points.

Reactions of the host received implantation of controlled release drug delivery devices are tissue/organ-dependent, species-dependent, and do happen in the same sequence as wound healing after tissue injury [45]. It has been shown that subcutaneous implantation of PLGA 75:25 drug-containing microspheres in rats provokes minimal inflammatory reaction with macrophages and giant cells, which disappears following biodegradation of the microspheres over a one-month period [46]. Furthermore, localized inflammation, and increased number of fibroblasts are characteristic features seen in fibrotic conditions such as keloids and hypertrophic scarring [47]. In our study, the statistically significant reduction in a total number of cells/field in PVA + KynA microspheres compared to the PVA alone group might partially be explained by the inhibitory effect of initially released KynA on fibroblast migration and proliferation as previously reported by our group [28]. In addition to KynA, localized increases in lactic acid concentration resulting from hydrolytic degradation of PLGA microspheres might also be a factor that explains the less total tissue cellularity in PVA + empty microspheres relative to PVA alone as well. This finding is in agreement with a study by Hasiao et al. that showed lactic acid inhibits the progression of the cell cycle at G1/S in the human keratinocyte cell line (HaCat) and induces apoptosis via caspase-dependent and caspase-independent pathways [48].

Another hallmark of fibrotic tissue is excessive collagen deposition following the initial repair process, which, in our study, was significantly reduced in samples containing KynA after 66 days. These findings might be explained by the delayed and gradual release of KynA from microspheres, as predicted by in vitro release kinetics and the residual KynA in retrieved implanted microspheres; and are in agreement with previously reported suppression of type-I collagen and fibronectin, and increase in MMP expression by KynA in primary dermal fibroblasts [28]. Furthermore, the number of myofibroblasts, as one of the essential role players in fibrotic conditions by contributing to ECM synthesis and wound contraction [49], was shown to be significantly reduced in the KynA microspheres group. Although PVA + empty microspheres group also showed less collagen deposition compared to the PVA alone after 66 days, the amount of collagen was significantly higher than the PVA + KynA microspheres group. The significant difference in collagen deposition between PVA + KynA and PVA + empty microspheres, despite the similarity in total cellularity and α-SMA expression, highlights the role of KynA in the prevention of fibrotic tissue formation.

While the reduction in collagen deposition in PVA + empty microspheres compared to PVA alone might partly be explained by a decreased number of myofibroblasts, further investigation is needed to elucidate the mechanism(s) involved in the partial anti-fibrotic effect of polymer-only microspheres.

## 5. Conclusions

We demonstrated that KynA can be encapsulated with high efficiency in PLGA + 17% MePEG diblock microspheres with suitable release kinetics, and the controlled delivery of KynA effectively reduces the fibrotic tissue formation in vivo. The feasibility of embedding KynA-loaded biodegradable polymer microspheres in surgical wounds before closure offers a promising method for the prevention and treatment of challenging fibrotic conditions.

## Figures and Tables

**Figure 1 pharmaceutics-14-01546-f001:**
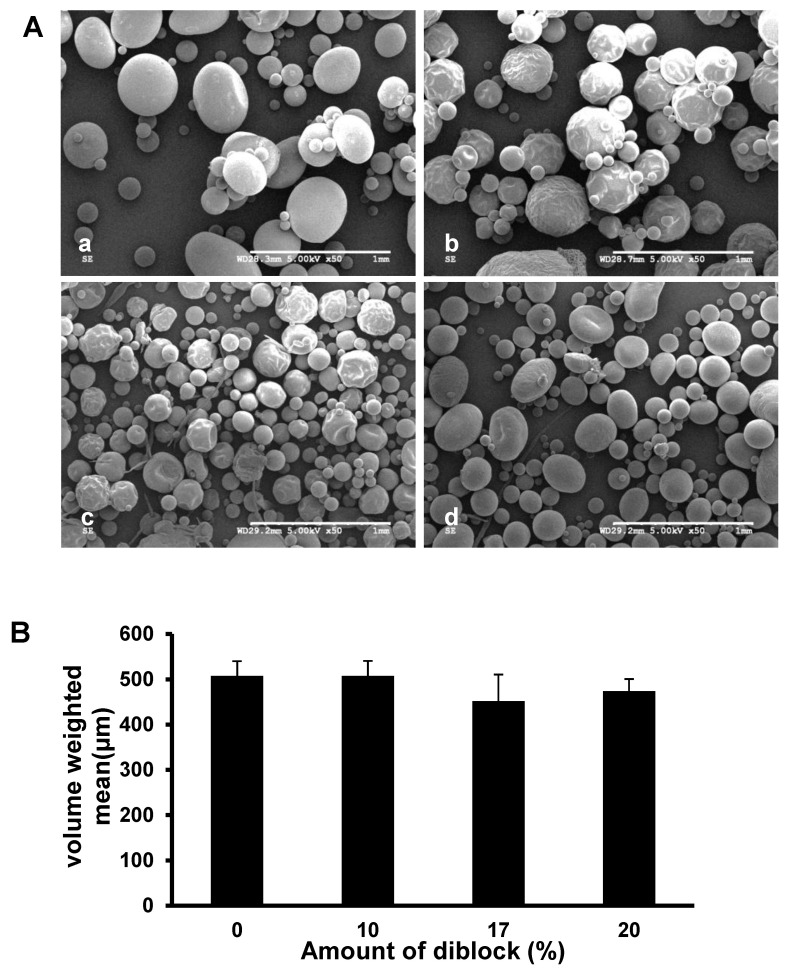
Scanning electron microscopy images and size distribution of different formulations of PLGA microspheres. (**A**) SEM images of KynA loaded PLGA microspheres fabricated without diblock (**a**) or with 10% (**b**), 17% (**c**), and 20% (**d**) MePEG- diblock. Scale bar: 1 mm. (**B**) Average size of the KynA loaded PLGA microspheres fabricated without or with 10%, 17%, and 20% MePEG-diblock. Particle size analysis of the microspheres was performed using laser diffraction particle size analyzer on four separate preparations of each formulation and the average of corresponding particle size was expressed as the volume weighted mean. Data represent the mean ± SD for *n* = 4.

**Figure 2 pharmaceutics-14-01546-f002:**
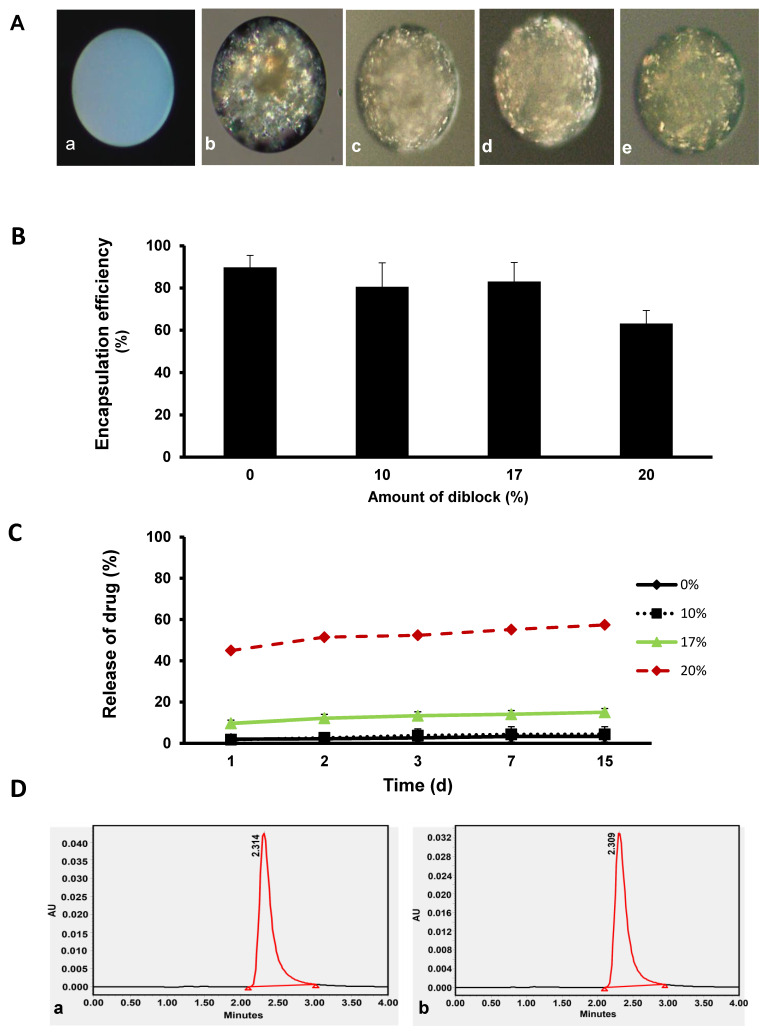
Encapsulation efficiency, in vitro drug release profile from the microspheres, and stability of KynA in the process of encapsulation (**A**) Polarized light microscopy images of empty (**a**), KynA loaded PLGA (**b**), PLGA+ 10% (**c**), PLGA + 17% (**d**), PLGA + 20% (**e**) MePEG-diblock microspheres, showed successful KynA encapsulation with all four formulations. (**B**) Quantitative analysis of the encapsulation efficiency of KynA loaded PLGA, PLGA + 10%, PLGA + 17%, and PLGA + 20% MePEG-diblock. Data represent the mean ± SD of independent batches (*n* = 4) for each formulation. (**C**) In vitro release profile of KynA from PLGA alone, PLGA+ 10%, 17%, and 20% MePEG-diblock microspheres. Cumulative release % was plotted vs. time (day) (mean ± SD, *n* = 4). (**D**) HPLC peak shape and retention time (t_R_) of the freshly prepared, standard solution of KynA (**a**) with KynA released from fabricated microspheres (**b**) showed identical results.

**Figure 3 pharmaceutics-14-01546-f003:**
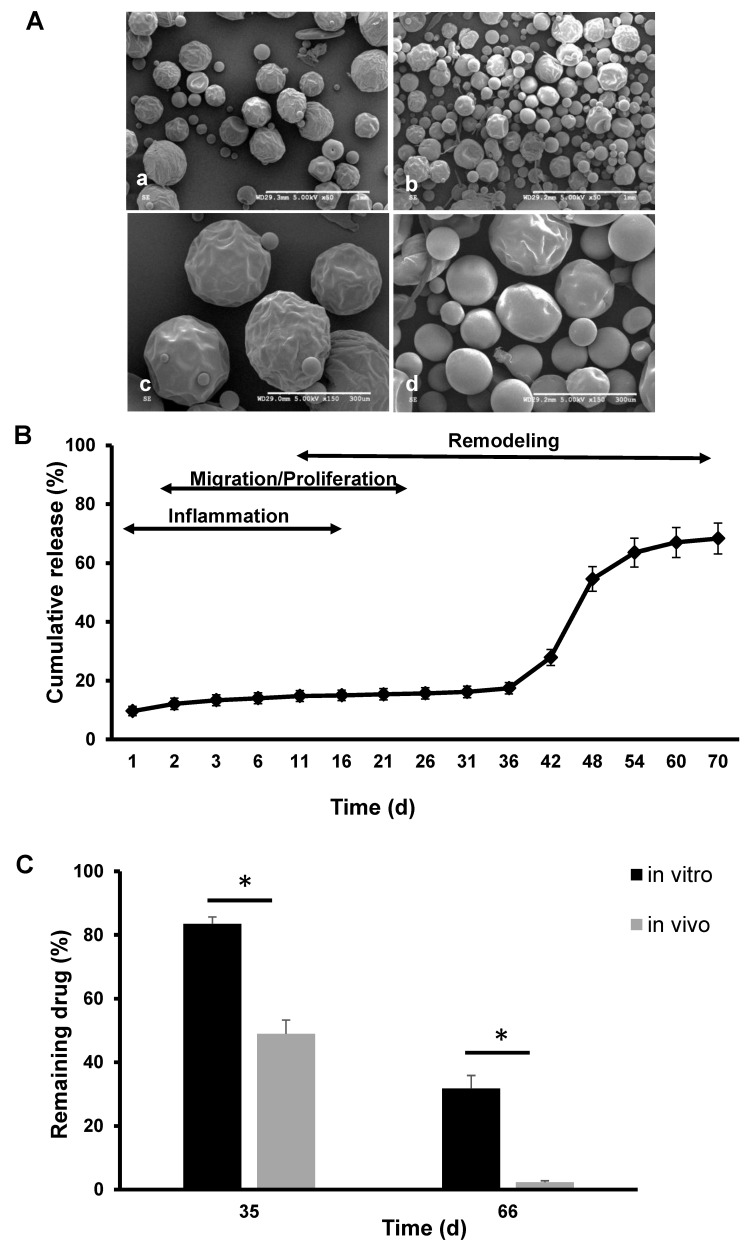
Scanning electron microscopy images of polymer only and KynA loaded microspheres, extended-release kinetics in vitro, and residual KynA inside microspheres in vitro and in vivo at different time points. (**A**) SEM images of empty PLGA + 17% MePEG-diblock (**a**,**c**) and KyA loaded PLGA + 17% MePEG-diblock microspheres (**b**,**d**). Scale bar: 1 mm (**a**,**b**), 300 μm (**c**,**d**). (**B**) Extended-release profile of KynA from PLGA + 17% MePEG-diblock microspheres in vitro up to 70 days, comparative to distinct phases of wound healing (schematic illustration). (**C**) Quantitative analysis of the residual KynA in microspheres at 35 and 66 days in vivo and in vitro. Data represent the mean ± SD for *n* = 4, * statistical significance, *p* < 0.05.

**Figure 4 pharmaceutics-14-01546-f004:**
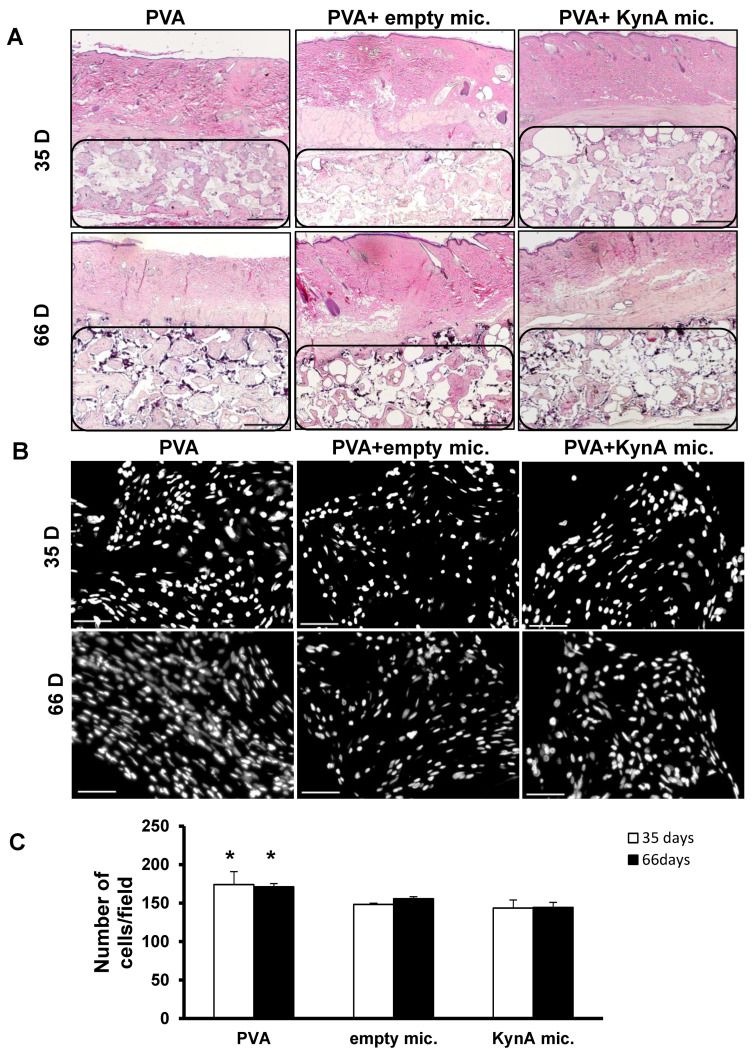
New tissue growth and total tissue cellularity inside PVA sponges of skin samples. (**A**) Representative sections of skin samples harvested after 35 and 66 days and stained with H&E. Granulation tissue grown inside of the PVA sponges (region marked by drawing) in all three groups: (i) PVA alone, (ii) PVA + empty microspheres, and (iii) PVA + KynA microspheres at ×20 magnification. Scale bar = 1 mm. (**B**) Tissue cellularity inside PVA sponges in sections of PVA alone, PVA + empty microspheres, and PVA + KynA microspheres at 35 and 66 days, stained with DAPI, at ×200 magnification. Scale bar = 50 μm. (**C**) Average number of total cells per field at day 35 (open bars) and day 66 (solid bars) in the PVA alone, PVA + empty, and PVA + KynA microspheres. Data represent the mean ± SD for *n* = 4, * Statistical significance, *p* < 0.01.

**Figure 5 pharmaceutics-14-01546-f005:**
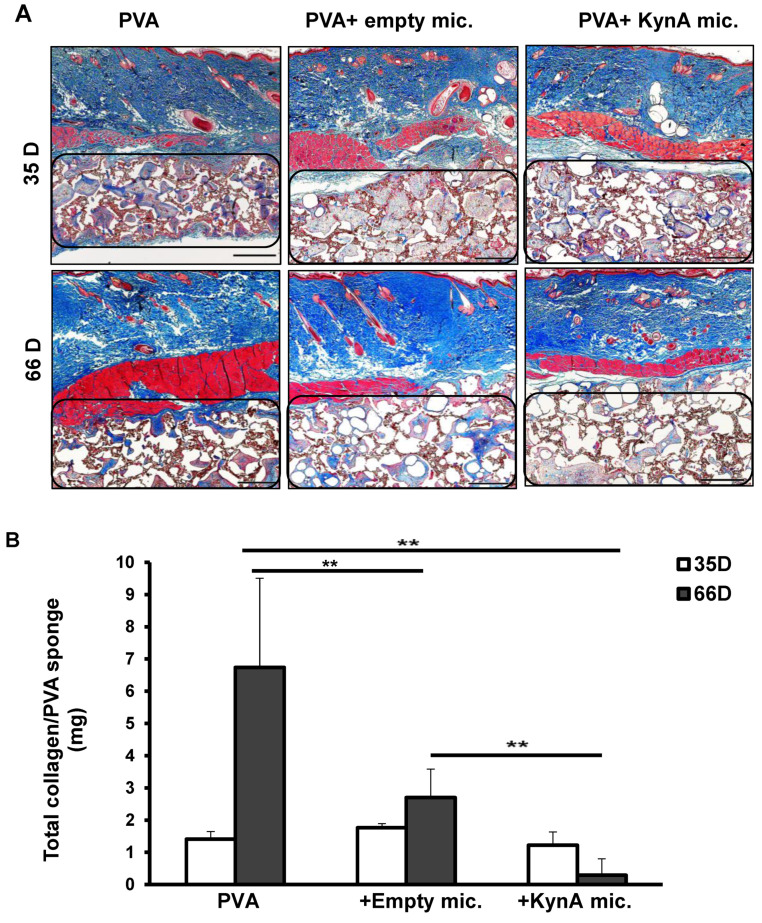
Biological effect of controlled release KynA on collagen deposition inside PVA sponges. (**A**) Masson’s trichrome staining of skin samples harvested at two time points (35 and 66 days) shows deposition of collagen (blue color) inside the PVA sponges (region marked by drawing) of three groups: (i) PVA alone, (ii) PVA + empty microspheres and (iii) PVA + KynA microspheres at ×20 magnification, scale bar: 1 mm. (**B**) Quantitative analysis of the collagen inside PVA sponges using hydroxyproline assay. The amount of collagen was measured in whole PVA sponges of three groups (i) PVA alone, (ii) PVA + empty microspheres, and (iii) PVA + KynA microspheres, harvested after 35 (open bars) and 66 days (solid bars). Data represent the mean ± SD, *n* = 4 for each group, ** statistical significance, *p* = 0.009.

**Figure 6 pharmaceutics-14-01546-f006:**
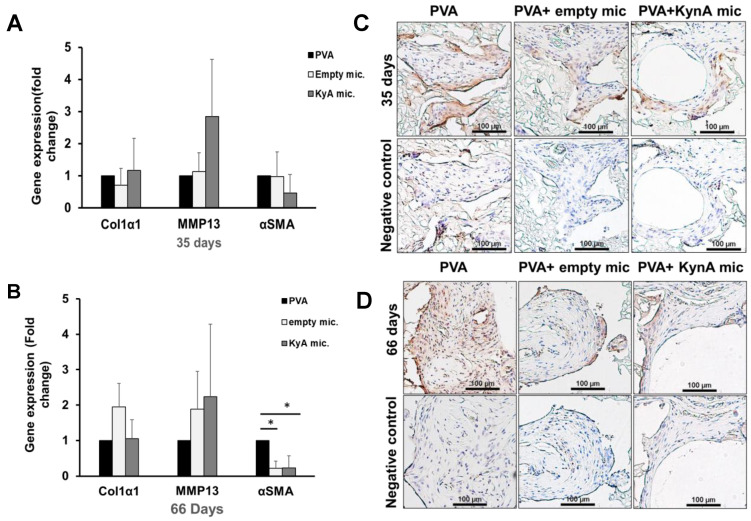
Expression of ECM components in tissue grown inside PVA sponges. (**A**,**B**) The levels of expression of Col1α1, MMP-13, and α-SMA genes in tissue grown inside PVA sponges of three groups: (i) PVA alone, (ii) PVA +empty microspheres, and (iii) PVA + KynA microspheres analyzed by qPCR at day 35 (**A**) and 66 (**B**). The results were normalized to β-actin as an internal control. Data represent as fold change relative to PVA alone group (mean ± SD, *n* = 3, * statistical significance, *p* = 0.009). (**C**,**D**) Immunohistochemical staining of skin samples from PVA alone, PVA + empty microspheres, and PVA + KynA microspheres groups for α-SMA. Cells expressing α-SMA in tissue grown inside PVA sponges at 35 (**C**) and 66 days (**D**), scale bar: 100 μm.

## Data Availability

Not applicable.

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
