# Peer review of "Localized Controlled Release of Kynurenic Acid Encapsulated in Synthetic Polymer Reduces Implant—Induced Dermal Fibrosis"

_pharmaceutics, 2022, doi:10.3390/pharmaceutics14081546_

Round 1

Reviewer 1 Report

The article is correctly framed in the SI theme, has a coherent presentation and is clearly detailed the innovative aspects related to the treatment and care of wounds.

A series of observations and suggestions from me are only on the form, format and word processing side, and are the following:

1. All figures must be framed in the same resolution, standardized, according to the requirements of the guide for authors.

2. Authors should also review the bibliographic indexes as indicated in the author's guide.

Reviewer 2 Report

The manuscript by Nabai et al. synthesized PLGA microspheres loaded with kynurenic acid for skin tissue repairs with the emphasis of reducing fibrosis formation. The authors performed in vitro assays as well as in vivo animal models to evaluate the release behaviors of kynurenic acid and collagen formation in tissues. Overall, the manuscript fits the scope of the Journal. However, there are some minor issues for the authors to address to improve the quality of the manuscript.

(1)  In section 3.1, the size distribution numbers need to have units.

(2)  Please improve the quality of Figure 2D, where the numbers and labels of x- and y-axes are hard to read.

(3)  It seems that the authors transition to PVA sponges using animal models. How is it related to the PLGA microspheres studies earlier? Please make the transition smoother.

(4)  Please justify the reason of using PVA sponges for the control groups.

(5)  Please double check for typos.

Reviewer 3 Report

  1. In the introduction, lines 32-42 do not seem to help the authors to understand the background. Since the author is trying to address the fibrosis associated with medical device implantation, the initial description in 32-42 are too broad to contribute to the paper.  
  2. References should be provided for statement in lines 75 and 76, 1) Journal of Controlled Release. 2019, pp 172–189.; 2) Angew. Chemie - Int. Ed. 2020, 59 (52), 23466–23470; 3) Chem. Soc. Rev. 2015, 44(17), 6161–6186; 4) Angew. Chemie - Int. Ed. 2020, 59 (26), 10456–10460 
  3. Figure 2D need to be replotted, it is hard to read the retention time on the x axis, also, the stacked graph will provide a better view of the comparison
  4. Why is the burst release so high for PLGA+20%? 
